# Novel Angiotensin-Converting Enzyme-Inhibitory Peptides Obtained from *Trichiurus lepturus*: Preparation, Identification and Potential Antihypertensive Mechanism

**DOI:** 10.3390/biom14050581

**Published:** 2024-05-15

**Authors:** Jiaming Cao, Boyuan Xiang, Baojie Dou, Jingfei Hu, Lei Zhang, Xinxin Kang, Mingsheng Lyu, Shujun Wang

**Affiliations:** 1Jiangsu Key Laboratory of Marine Bioresources and Environment/Jiangsu Key Laboratory of Marine Biotechnology, Jiangsu Ocean University, Lianyungang 222005, China; jmcao@jou.edu.cn (J.C.); xby@jou.edu.cn (B.X.); bjdou@jou.edu.cn (B.D.); jfhu123@jou.edu.cn (J.H.); leizhang@jou.edu.cn (L.Z.); kangxinxin@jou.edu.cn (X.K.); mslyu@jou.edu.cn (M.L.); 2Co-Innovation Center of Jiangsu Marine Bio-Industry Technology, Jiangsu Ocean University, Lianyungang 222005, China

**Keywords:** *Trichiurus lepturus*, ACE inhibitory peptides, in silico, molecular docking

## Abstract

Peptides possessing antihypertensive attributes via inhibiting the angiotensin-converting enzyme (ACE) were derived through the enzymatic degradation of *Trichiurus lepturus* (ribbonfish) using alkaline protease. The resulting mixture underwent filtration using centrifugation, ultrafiltration tubes, and Sephadex G-25 gels. Peptides exhibiting ACE-inhibitory properties and DPPH free-radical-scavenging abilities were isolated and subsequently purified via LC/MS-MS, leading to the identification of over 100 peptide components. In silico screening yielded five ACE inhibitory peptides: FAGDDAPR, QGPIGPR, IFPRNPP, AGFAGDDAPR, and GPTGPAGPR. Among these, IFPRNPP and AGFAGDDAPR were found to be allergenic, while FAGDDAPRR, QGPIGPR, and GPTGPAGP showed good ACE-inhibitory effects. IC_50_ values for the latter peptides were obtained from HUVEC cells: FAGDDAPRR (IC_50_ = 262.98 μM), QGPIGPR (IC_50_ = 81.09 μM), and GPTGPAGP (IC_50_ = 168.11 μM). Peptide constituents derived from ribbonfish proteins effectively modulated ACE activity, thus underscoring their therapeutic potential. Molecular docking and modeling corroborated these findings, emphasizing the utility of functional foods as a promising avenue for the treatment and prevention of hypertension, with potential ancillary health benefits and applications as substitutes for synthetic drugs.

## 1. Introduction

Angiotensin-converting enzyme is considered a major determinant of the control of blood pressure and cardiovascular-disease-related morbidity as it is responsible for releasing angiotensin-II, a potent vasoconstrictor, by cleaving angiotensin-I while simultaneously allowing renin inactivation of the vasomotor factor bradykinin in the angiotensin system, ultimately leading to enhanced vasoconstriction, which leads to hypertension [1]. Bioactive peptides have attracted a great deal of interest among researchers because of their great potential as functional food ingredients. Collagen peptides from various sources have been used in various products, such as food, cosmetics, pharmaceutical/biomedical products, and nutritional products [2]. Collagen sourced from aquatic animals was more susceptible to enzymatic hydrolysis and thus more efficient for preparing bioactive peptides compared to mammalian collagen. Therefore, it is considered one of the hydrolysates with superior nutritional properties.

Edible aquatic animals, such as seafood and their by-products, possess substantial potential for functional food applications and other dietary interventions [3]. In addition, aquatic peptides can be tolerated by most people with religious beliefs [4]. However, ACE-inhibitory peptides derived from ribbonfish are currently less studied. Food processing often yields a substantial number of by-products or waste materials that possess the potential for producing bioactive peptides. In our previous work, we optimized the process parameters for generating bioactive peptides from *Trichiurus lepturus* (ribbonfish). The objective of this study was to purify bioactive peptides from ribbonfish protein hydrolysate and assess their antihypertensive properties.

## 2. Materials and Methods

### 2.1. Materials

Ribbonfish (Jiadefu Supermarket, Lianyungang, China), Angiotensin I-converting enzyme (ACE) with an activity of 0.25 U/mg sourced from rabbit lungs, captopril, and 2,2-diphenyl-1-picrylhydrazyl (DPPH) were purchased from Sigma-Aldrich (St. Louis, MO, USA). Alcalase, trypsin, pepsin, papain, and neutral protease were purchased from Aladdin Biochemical Technology Co., Ltd., Shanghai, China. All remaining chemicals utilized in this study were of either chromatographic or analytical grade.

### 2.2. Methods

#### 2.2.1. Preparation of Enzyme Digests

Ribbonfish protein was prepared using the method outlined by Priscilla et al. [5] with minor adjustments. Hydrolysis was carried out at a substrate concentration of 100 g/L in 1 L conical flasks. Incubation was carried out at various optimal temperatures and pH levels for protease activity for 4 h at 100 rpm. The reaction was terminated via heating at 85 °C for 20 min. Subsequently, the hydrolysate was centrifuged at 8000× *g* for 20 min at 10 °C, followed by freeze-drying and storage at 4 °C until further use.

#### 2.2.2. Separation of Peptides via Ultrafiltration

The products displaying the highest in vitro ACE-inhibitory activity following alkaline protease hydrolysis were separated using Amicon^®^ Ultra 15 mL centrifugal filters comprising molecular weight (MW) cut-off membranes, specifically with respect to 10 kDa and 3 kDa (EMD Millipore; Billerica, MA, USA). Three filtrates were obtained with MW < 3 kDa, 3 < MW < 10 kDa, and MW > 10 kDa, respectively. These filtrates were subsequently freeze-dried for the evaluation of their biological activity.

#### 2.2.3. Separation of Peptides via Gel Chromatography

The ultrafiltrate (50 mg) with potent antioxidant and ACE-inhibitory properties was subjected to screening and dissolution in 1 mL of distilled water before separation via a Sephadex G-25 (Sigma Aldrich, St. Louis, MO, USA) column measuring 2.5 × 100 cm. The elution buffer was maintained at a pH of 6.5, concentration of 50 mM of Tris-HCl, and flow rate of 1.5 mL/min, with detection performed at 280 nm. The fractions were individually collected based on absorption peaks, and subsequent lyophilization was carried out to confirm ACE-inhibitory activity.

#### 2.2.4. Peptide Sequencing and Identification via LC-MS/MS

After desalination, the peptide samples were dried through centrifugation and then re-dissolved in 100 μL of Nano-LC mobile phase A, which is composed of 0.1% formic acid and water. The reconstituted samples were then loaded onto a nano Viper C18 pre-column (3 μm, 100 Å) with an up-sampled volume of 2 μL, followed by a 20 μL volumetric rinse for desalination. The liquid phase system used was an Easy nL C 1200 nL (Thermo Fisher, Waltham, MA, USA). The samples were desalted on the pre-column before their separation on the analytical column, which was a C18 reversed-phase column (Acclaim PepMap RSLC, Thermo Fisher, Waltham, MA, USA, 75 μm × 25 cm C18-2 μm 100 Å). The gradient employed for the experiments was a 30 min gradient of mobile phase B (80% acetonitrile, 0.1% formic acid) increased from 5% to 38%. The mass spectra were obtained using a Thermo Fisher Q Exactive system (Thermo Fisher, USA) coupled with a nanolitre spray Nano Flex ion source (Thermo Fisher, USA) with a spray voltage of 1.9 kV and an ion transfer tube heated to 275 °C. The mass spectra were acquired in an information-dependent mode. Mass spectrometry was performed using the information-dependent acquisition mode (DDA, Data Dependent Analysis). The primary mass spectrometry scanning resolution was set to 70,000, covering a scanning range of 100–1500 *m*/*z*, with a maximum injection time of 100 ms. Within each data-dependent acquisition (DDA) cycle, a maximum of 20 secondary scans with charges ranging from 1+ to 3+ were obtained, with a maximum injection time of 50 ms for the secondary mass spectrometry ions. The collision energy in the collision chamber for all precursor ions was fixed at 28 eV using high-energy collision-induced dissociation (HCD), and a dynamic exclusion period of 6 s was employed. The acquired raw spectrum files from mass spectrometry were processed and analyzed using PEAKS Studio 8.5 software developed by Bioinformatics Solutions Inc. (Waterloo, Canada).

#### 2.2.5. Screening of Peptides

Identified peptides were analyzed in the peptide Ranker database accessible via the following link: http://distilldeep.ucd.ie/PeptideRanker/ (Accessed 29 November 2023) Peptides with bioactive potential were distinguished using a threshold of 0.5 [6]. Active peptides were assessed using the AHTpin database (http://crdd.osdd.net/raghava/ahtpin/index.php (Accessed 29 November 2023)) and identified as antihypertensive peptides if their scores exceeded 0 [7]. Subsequently, antihypertensive peptide activity was determined using the BIOPEP database (https://biochemia.uwm.edu.pl/biopep-uwm/ (Accessed 29 November 2023)) by utilizing parameter B with a cut-off of 0.03 to estimate potential antihypertensive activity [8]. To predict the peptides’ hypoallergenicity, the AllerTOP database (https://www.ddg-pharmfac.net/allertop/index.html (Accessed 29 November 2023)) was employed [9]. The stability of non-hazardous peptides in the gastrointestinal tract was determined using the HLP database (https://webs.iiitd.edu.in/raghava/hlp/ (Accessed 29 November 2023)) to identify peptides that are tolerant to digestive fluids [10]. The ToxinPred database (https://webs.iiitd.edu.in/raghava/toxinpred/index.html (Accessed 30 November 2023)) was used to predict the prospective toxicity and separation locations of the selected peptides [11]. Physical properties were determined using the AHTpin database (http://crdd.osdd.net/raghava/ahtpin/index.php (Accessed 30 November 2023)), while plasma stability was assessed through the PlifePred database (https://webs.iiitd.edu.in/raghava/plifepred/batch.php (Accessed 30 November 2023)) [12].

#### 2.2.6. ACE-Inhibitory Property

ACE-inhibitory activity was assessed by following the experimental procedure reported by Dou et al. [13], with slight modifications. A solution of borate-buffered HEPES (50 mM, pH 8.3) was dispensed into a 96-well plate, followed by the addition of samples (100 µL), ACE (50 µL, 0.1 U/mL, prepared from 50 mM borate buffer pH 8.3), and FAPGG(N-[3-(2-Furyl)acryloyl]-Phe-Gly-Gly) (50 µL, 1 mM) into each well. The negative control group was substituted with 100 µL of borate-buffered HEPES. The initial absorbance values of a_1_ and b_1_ for both the blank and sample groups were then measured at 340 nm (Thermo Fisher, Waltham, MA, USA). The sample group was then incubated at 37 °C for 30 min, and subsequently, the absorbance values were read again at 340 nm (a_2_, b_2_).
(1)$$ACE\;inhibition\;rate=\frac{A-B}{A}\times 100\%$$
where *A* = a_1_ − a_2_, and *B* = b_1_ − b_2_. The initial absorbance of the blank: a_1_—blank; b_1_—sample. a_2_: the absorbance values after incubation at 37 °C for 30 min. a_2_: blank; b_2_: samples.

#### 2.2.7. DPPH Scavenging Activity

After mixing 0.1 mL of the sample with 0.1 mL of 0.1 mM DPPH solution, the mixture was placed in a dark area. Following a 30 min incubation period (Thermo Fisher, Waltham, MA, USA), the OD _517_nm value of the mixture was measured. The DPPH scavenging activity was then calculated using the following formula:(2)$$DPPH\;clearance=(1-\frac{a_{x}-a_{x0}}{a_{0}})\times 100\%$$
where *a_x_* represents the absorbance of the peptide–DPPH reaction, *a_x_*_0_ indicates the absorbance of the peptide–ethanol reaction, and *a*_0_ refers to the absorbance of the DPPH–water reaction.

#### 2.2.8. Molecular Docking of Peptides with ACE

The three-dimensional structure of human ACE (PDB ID 1O8a) was obtained from Protein Data Bank (https://www.rcsb.org/) [14]. Pymol 2.5 software was used to optimize 1O8a for the removal of water molecules and small-molecule ligands. The peptide structures were constructed with Discovery Studio 2019 with the addition of the CHARMm force field to all peptides. Subsequently, the peptides underwent energy minimization utilizing the Smart Minimizer algorithm, with a maximum of 2000 steps and an RMS gradient value set to 0.01. Molecular docking was carried out using 1O8a as the receptor and the peptides as the ligand. The binding energy was calculated, and the results were then visualized using PyMol 2.5 software. The affinity value (kcal/mol) represents the binding capacity. The lower the binding capacity, the more stable the ligand–receptor binding.

#### 2.2.9. In Vitro Simulation of Oral Gastrointestinal Digestion

The in vitro digestion process was utilized [15]. A quantity of 1 mg of the sample was dissolved in 1 mL of water and preheated in a water bath at 37 °C. Then, it was mixed with 1 mL of artificial saliva whose pH was adjusted to pH 6.8. The mixture was then agitated at 100 rpm for 1 min at 37 °C. Subsequently, the pH was adjusted to pH 2.0 using 1 M HCl and added in equal amounts to the simulated gastric digest. The mixture was then incubated at 37 °C for 90 min. This mixture was then combined with the simulated enteric digest in a 1:1 ratio and incubated at −4 °C for 120 min. After the initial gastric digestion, the pH was adjusted to 7.5 using 1 M of NaHCO_3_. For the subsequent stage of digestion, the pH was again adjusted to 7.5 using 1 of M NaHCO_3_, and the samples were added to the simulated intestinal digest in a 1:1 ratio. This mixture was then incubated at 37 °C for 120 min. Following each stage of digestion, the samples were collected and centrifuged at 8000× *g* for 10 min at 4 °C. The supernatant was then extracted and stored at −80 °C for later use [16].

#### 2.2.10. Cell Culture

Cell culture procedures were carried out according to the literature. Human umbilical vein endothelial cells (HUVEC cells) were purchased from Bena Bioengineering Company (Wuhan, China) and cultured in high-glucose DMEM medium supplemented with 10% fetal calf serum and 100 units/mL of penicillin at 37 °C and 100 μg/mL of streptomycin. For in vitro experiments, cells were divided into control, peptide (250 μg/mL), and peptide (500 μg/mL) groups.

#### 2.2.11. Cell Viability Analysis

HUVEC cells at a density of 3 × 10^5^ cells/mL were seeded into 96-well plates and then incubated at 37 °C for 24 h. Thereafter 100 μL of Cell Counting Kit-8 (CCK-8) solution (Beyotime, Beijing, China) was added to each well and incubated at 37 °C for 4 h. The optical density was then measured at 450 nm using a microplate reader (Thermo Fisher, Waltham, MA, USA). 

#### 2.2.12. Determination of ACE Inhibition Mode

Different concentrations of FAPGG were used to measure ACE-inhibitory activity as described in Section 2.2. The Michaelis–Menten constant (K_m_), maximum reaction rate (V_max_), and inhibitor constant (K_i_) of the peptides were determined using Lineweaver–Burke plots according to the Michaelis–Menten equation [17].

#### 2.2.13. Statistical Analysis

All experiments were conducted in triplicate, and the results are expressed as means ± standard deviations (SDs). The data analysis was performed using SPSS software (usedIBM SPSS Statistics 23) (Chicago, IL, USA), identifying significant differences between results (*p* < 0.05) using one-way ANOVA and Duncan’s method. Duncan’s multiple extreme variance test was also employed to compare the mean values and determine significant differences between the results (*p* < 0.05).

## 3. Results

### 3.1. Preparation of Enzyme Digests

The ACE-inhibiting activities of the neutral-protease- [18], trypsin-, papain-, pepsin-, and alkaline-protease-treated samples are demonstrated in Figure 1. The hydrolysate treated with alkaline protease displayed the greatest ACE-inhibitive activity, amounting to 86.7% [19]. Proteases have selectivity towards the amino acid compositions of peptide bonds when they hydrolyze substrates. Therefore, screening enzymes were beneficial for improving the biological activity of the hydrolysates [20].

### 3.2. Separation of Peptides through Ultrafiltration

Alkaline protease hydrolysates are categorized based on their molecular weights: >10 kDa, 3~10 kDa, and <3 kDa. Previous studies have indicated that peptides possessing low molecular weight exhibit greater inhibitory activity against ACE [21]. Hence, it can be inferred that the <3 kDa fraction may display stronger inhibitory activity against ACE. Another supposition suggests that there exists a direct correlation between the hydrophobicity of peptides and ACE-inhibitory activity. Hence, we isolated the <3 kDa fraction through G25 gel chromatography and acquired four fractions that were lyophilized for further examination (Figure 2a). Amongst all the fractions (Figure 2b), only fraction 4 (F4) demonstrated an ACE-inhibitory activity of 0. This could suggest that the sample had already gone through the G25 dextran gel column, which was the endpoint of fraction 3. Fractions 1 (F1) and 3 (F3) demonstrated similar inhibitory activity, which is likely due to the lower concentration of solutes in the collection solution at the beginning and end of the process. Peptides with molecular weights greater or less than the weight of fraction 2 were less abundant in fractions less than 3 kDa. The peptides’ greater hydrophobicity increased their effectiveness in inhibiting ACE compared to their counterparts. Hence, F2 had the highest ACE-inhibitory activity among all four sites. We hypothesize that specific structural features of these peptides favor ACE inhibition [22,23].

### 3.3. LC/MS-MS Chromatography and Peptide Synthesis

LC-MS/MS peptide sequence analysis identified over 100 peptides in F2. The three peptides are displayed based on the screening results in Section 3.4, and they are FAGDDAPR, QGPIGPR, and GPTGPAGPR (Figure 3). This text adheres to the principles of clarity, objectivity, and logical structure in line with academic conventions. The diverse bioactive functions of peptides stem from the composition and sequence of a peptide as well as the size and type of amino acids at the amino or carboxy terminus [24]. These factors impact the solubility and hydrophobicity of peptides, directly influencing their activity and ability to be absorbed by the body. A website was used to predict that the secondary structures of six peptides were randomly coiled. Ossama Daoui et al. demonstrated a strong correlation between 2D and 3D structural properties and inhibitory activity [25]. Jarosław Ruczyński et al. reported that the α-helical structure of a peptide had a significant impact on its interaction with negatively charged membranes. They also found a strong correlation between the helix content of a peptide and antimicrobial activity [26]. This study demonstrated a correlation between the structure of peptides and their biological activity. However, David Salehi et al. indicated that there was no correlation between peptide morphology and cellular uptake properties [27]. In the cited study, secondary structure related to the peptides’ antioxidant activity [28]. A greater proportion of irregular curls indicates a looser peptide structure, leading to better exposure of the active site and thereby enhancing receptor binding [29].

### 3.4. In Silico Sieve

Table 1 presents the corresponding in silico results (Appendix A is presented in Appendix A). PeptideRanker, for which a neural network model was leveraged, was consistently employed to forecast peptide bioactivity. Peptides with a score exceeding 0.5 were deemed biologically active. Subsequent to the initial screening, 43 peptides were chosen based on their sequence length. Effective ACE-inhibitory peptides (ACEIPs) typically comprise 2–12 amino acids. Hence, we conducted further analysis on peptides with sequence lengths of up to 10 to assess their anti-hypertensive properties. Using AHTPDB, an additional bioinformatics platform utilizing support vector machine calculations, peptides scoring above 0 were designated as antihypertensive peptides. Furthermore, the selected peptides underwent evaluation via the BIOPEP-UWM database using parameters A (the frequency of peptide occurrence in the protein sequence) and B (ACE inhibition potential in μM). A crucial screening threshold for highly active ACEIPs was established, namely, a parameter B value exceeding 0.03. Subsequently, the five peptides identified through screening with parameter B were assessed using the AllerTOP and HLP databases to forecast their sensitization and intestinal stability. The peptides were predicted to possess high ACE-inhibitory activity, be non-sensitizing, and exhibit stability in the gut. Ensuring the safety of food is paramount. Thus, the ToxinPred database was utilized to predict the toxicity of five peptides, namely, FAGDDAPR, QGPIGPR, IFPRNPP, AGFAGDDAPR, and GPTGPAGPR, all of which were found to be non-toxic. While calculations can swiftly and cost-effectively assess peptide safety, validation through cellular and preclinical experiments remains necessary. The peptides’ molecular weights were all below 1 kDa, facilitating rapid absorption by the gastrointestinal tract and subsequent development of their biological activities. The predicted half-lives of the peptides in plasma ranged from 775.81 to 878.81 s. Additionally, these peptides exhibited efficacy in treating liver and kidney conditions. The physical properties of the target peptides, including hydrophobicity, hydrophilicity, amphiphilicity, and isoelectric point, serve as crucial indicators of their suitability in food processing. The hydrophobicity of the studied ACEIPs correlated with their hydrophobic amino acid content [30]. Notably, QGPIGPR displayed a high amphiphilicity value of 0.53, suggesting its potential for use in both polar and nonpolar food systems. Moreover, the pH value of FBSSH was determined to be 5.0, and the isoelectric points of the five peptides (all greater than 5) were alkaline, indicating their relative stability in food systems.

### 3.5. Molecular Docking of Peptides with ACE

Molecular docking is a technique that mimics small-molecule ligand–receptor biomolecules. It is a technique used to simulate the interaction between a small-molecule ligand and a receptor biomolecule, helping to form a binding site on the surface of the protein after the complex and to determine the binding mode and affinity of the small molecule to the receptor [31]. Table 2 displays the docking sites of the molecular docking between peptides and ACE. Figure 4 provides a 3D model map of docking.

### 3.6. In Vitro Simulation of Oral Gastrointestinal Digestion

Figure 5 shows that at all stages of digestion, the DPPH radical-scavenging rate of all three peptides decreased with digestion, using glutathione as a reference, and in the initial stage, the highest DPPH radical-scavenging rate of the three peptides was exhibited by GPTGPAGPR, whose DPPH radical-scavenging rate was 10.32% lower than that of glutathione, and the lowest was FAGDDAPR, whose DPPH radical-scavenging rate was 18.76% lower than that of glutathione. Th is was 32% lower than that of glutathione, and the lowest was exhibited by FAGDDAPR, whose DPPH radical-scavenging rate was 18.76% lower than the DPPH radical-scavenging rate of glutathione. The low DPPH free-radical-scavenging activity may be due to the fact that alkalase is a serine protease and endopeptidase with serine as the main catalytic site. Thus, alkaline phosphatase produces peptides with no significant activity. In contrast, hydrolysis by pepsin significantly increased DPPH radical-scavenging activity. This is due to the fact that pepsin cleaves the peptide bonds of adjacent aromatic amino acids such as phenylalanine, tryptophan, and tyrosine, and it has been shown that some dipeptides and tripeptides containing aromatic amino acid residues (tryptophan, tyrosine, etc.) exhibit higher antioxidant activity by providing hydrogen ions [22]. Using captopril as a reference, the ACE-inhibitory activity of the three peptides also decreased with digestion, but the difference was that, initially, the peptide with the lowest DPPH radical-scavenging rate of the three peptides, FAGDDAPR, had the highest ACE-inhibitory activity, and the ACE-inhibitory activity of FAGDDAPR was 31.17% lower than the ACE-inhibitory activity of captopril.

### 3.7. Determination of ACE Inhibition Mode

To evaluate whether the peptides inhibited ACE in a substrate-competitive manner, Lineweaver–Burk plot analysis was conducted (Figure 6). This involved examining the inhibition of ACE at varying peptide concentrations alongside different substrate concentrations (0.5, 1 mg/mL). The data analysis was carried out using the Michaelis–Menten equation program developed in Origin. ACE inhibitors interact primarily with key residues in the ACE catalytic site and act as competitive inhibitors [32]. Similarly, a wide range of non-competitive ACE inhibitors have been identified [33,34]. Non-ACE competitive inhibitors may inhibit ACEs by inducing conformational changes [35]. Further explanation is required regarding the specific mechanism of inhibition. Additionally, there have been reports of ACE-inhibitory peptides with a mixed mode of inhibition [36]. This suggests that these peptides may interact with ACE in a multifaceted manner, possibly involving both competitive and non-competitive inhibition pathways. In Figure 6, the Vmax values have decreased, and the Km values have increased, suggesting the peptide might interact with both the active site (competitive) and the variable structural site (non-competitive) of ACE, similar to LEPWR [17]. Further investigation is necessary to fully understand these peptides’ complex mechanisms of action.

### 3.8. Cell Viability Analysis

According to the literature [6], ACE-inhibitory peptides with an IC50 between about 0.32 and 1000 μM have the potential to lower blood pressure. The literature suggests a possible relationship between IC_50_ and the amount of hydrogen bond formation [37]. For example, VGINYW (IC_50_ 15.1 μM) from α-lactalbumin and WAGP (IC_50_ 140.70 μM) from carnosine protein have demonstrated antihypertensive effects. We therefore hypothesized that the peptides from FAGDDAPR, QGPIGPR, and GPTGPAGPR may have antihypertensive effects. The IC_50_ values of FAGDDAPR, QGPIGPR, and GPTGPAGPR are shown in Figure 7. The C-terminal structural domain is deemed crucial for blood pressure regulation. Consequently, hydrophobic peptides tend to exhibit a stronger affinity for ACE due to the hydrophobic environment of the ACE C-terminal structural domain. Numerous studies have reported that peptides with higher ACE-inhibitory potency often possess an aromatic amino acid (such as Trp, Tyr, or Phe) at the C-terminus, along with positively charged amino acids (such as Lys or Arg) at either the C- or N-terminus, particularly if the latter comprises hydrophobic aliphatic branched chain amino acids (such as Ile, Ala, Met, and Leu). This observation aligns with previous research findings (regarding Arg, Lys, and His). Prolonged use of peptides targeting ACE might lead to the development of resistance, diminishing their effectiveness over time. Furthermore, the antihypertensive effects of peptides need to be assessed. Blood pressure measurements, cardiac function assessments, and biochemical analyses could be performed to evaluate the efficacy and safety of peptides in vivo.

## 4. Discussion

The hydrolysis process of ribbonfish was meticulously optimized using five proteases to obtain hydrolysates with the highest ACE-inhibitory activity. Subsequently, peptides from the ribbonfish hydrolysates were isolated and identified through a combination of filtration and gel filtration chromatography techniques. Among the identified peptides, FAGDDAPRR, QGPIGPR, and GPTGPAGP exhibited notable antioxidant activity and robust ACE-inhibitory activity, demonstrating IC_50_ values of 262.98 μM, 81.09 μM, and 168.11 μM, respectively. Similarly, Dong et al. used alkaline protease to hydrolyze tilapia (*Oreochromis niloticus*) skin, and the oligopeptide VGLFPSRSF was obtained. Its ACE-inhibitory IC_50_ value was 61.43 μM [38]. Li et al. identified peptides from *Pinctada fucata* with potential ACE-inhibitory activity, and the IC50 value of the FRVW was 18.34 μM [39]. Wang et al. hydrolyzed Pacific Saury (*Cololabis saira*) via neutral protease to select ACE-inhibitory activity peptides, and they found the peptide LEPWR, for which the ACE-inhibitory IC50 value was 99.5 μM [17]. Based on the IC50 of ACE inhibition, our results were slightly higher. 

Analysis of binding sites between ACE and peptides and inhibition kinetics would lead to a better understanding of the modes of activity of peptides [40,41,42]. In addition, the C-terminal amino acid of peptides could affect their ACE-inhibitory activity significantly, and the hydrophobicity and three-dimensional structure of the C-terminal amino acids are crucial factors for their ACE-inhibitory activity [13,41,43]. Moreover, gastrointestinal digestion results could determine whether peptides are affected by enzymes of the gastrointestinal tract [44,45,46].

Our study underscores the potential of peptide synthesis and molecular dynamics simulations as valuable tools for further exploration and understanding of peptides’ mechanisms of action. Further in vivo studies are warranted to validate the contribution of specific amino acids to enzyme inhibition and evaluate the health benefits and metabolic effects of bioactive peptides derived from ribbonfish. This study provides a technical and theoretical basis for the potential use of ribbonfish hydrolysate as a functional food to combat hypertension. The next step is to evaluate them for their stability, bioavailability, immunogenicity, specificity, and off-target effects in suitable preclinical models. 

## Figures and Tables

**Figure 1 biomolecules-14-00581-f001:**
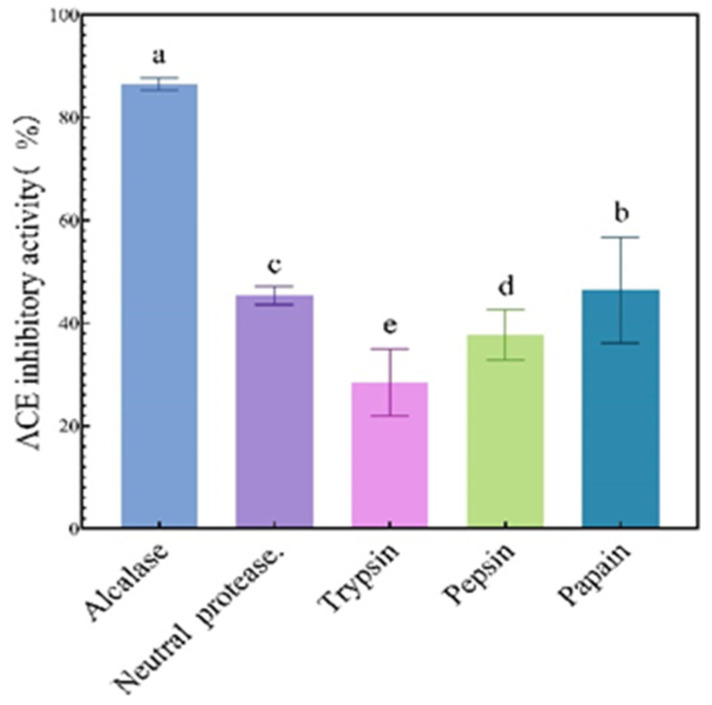
The products derived from treating ribbonfish with various types of proteases demonstrate ACE-inhibitory activity. The different letters represent the statistically significant difference (*p* < 0.05).

**Figure 2 biomolecules-14-00581-f002:**
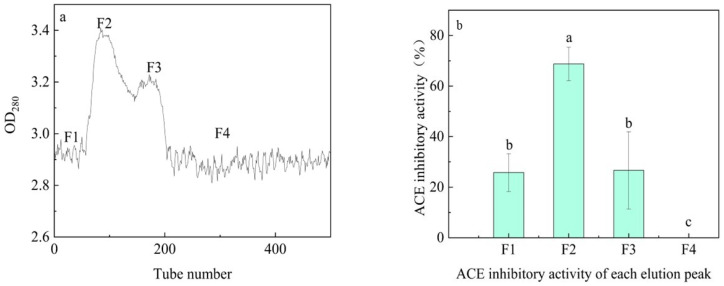
Purification of the various enzymatic products, (**a**) showing the elution profile in G25 and (**b**) presenting the ACE-inhibitory activity of each elution peak. Statistically significant differences are denoted by different letters.

**Figure 3 biomolecules-14-00581-f003:**
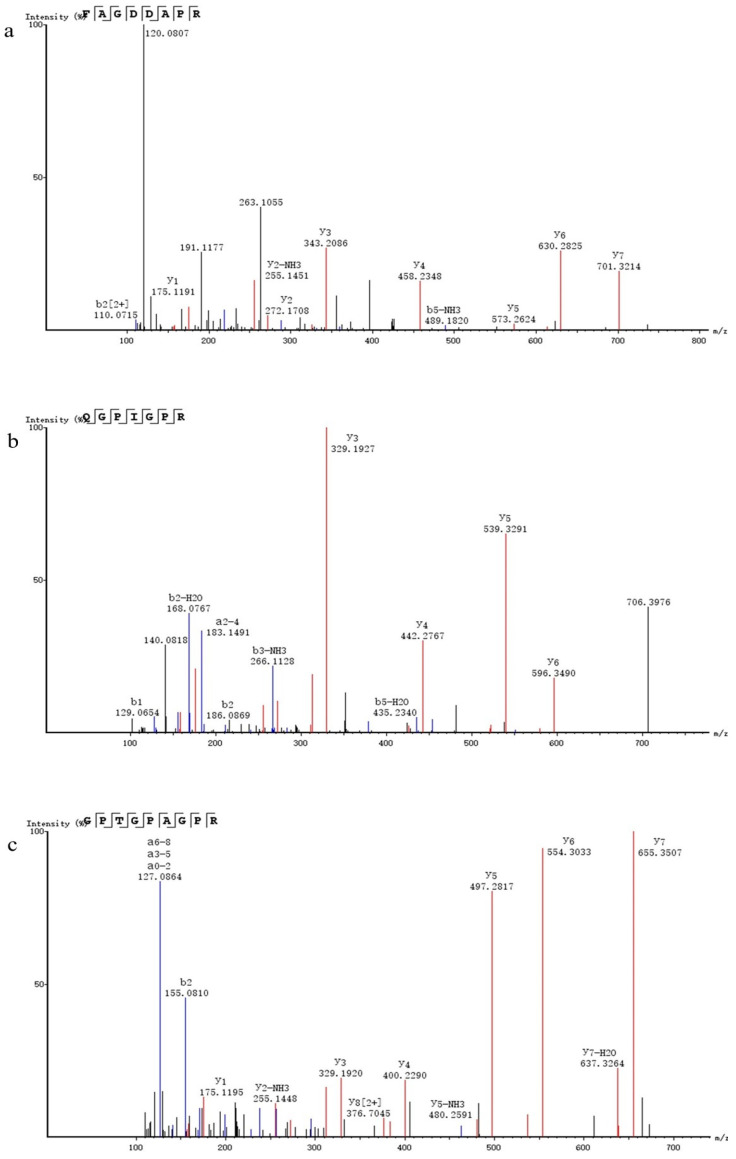
Analysis of purified and synthetic peptides using mass spectrometry. (**a**) FAGDDAPR, (**b**) QGPIGPR, and (**c**) GPTGPAGPR.

**Figure 4 biomolecules-14-00581-f004:**
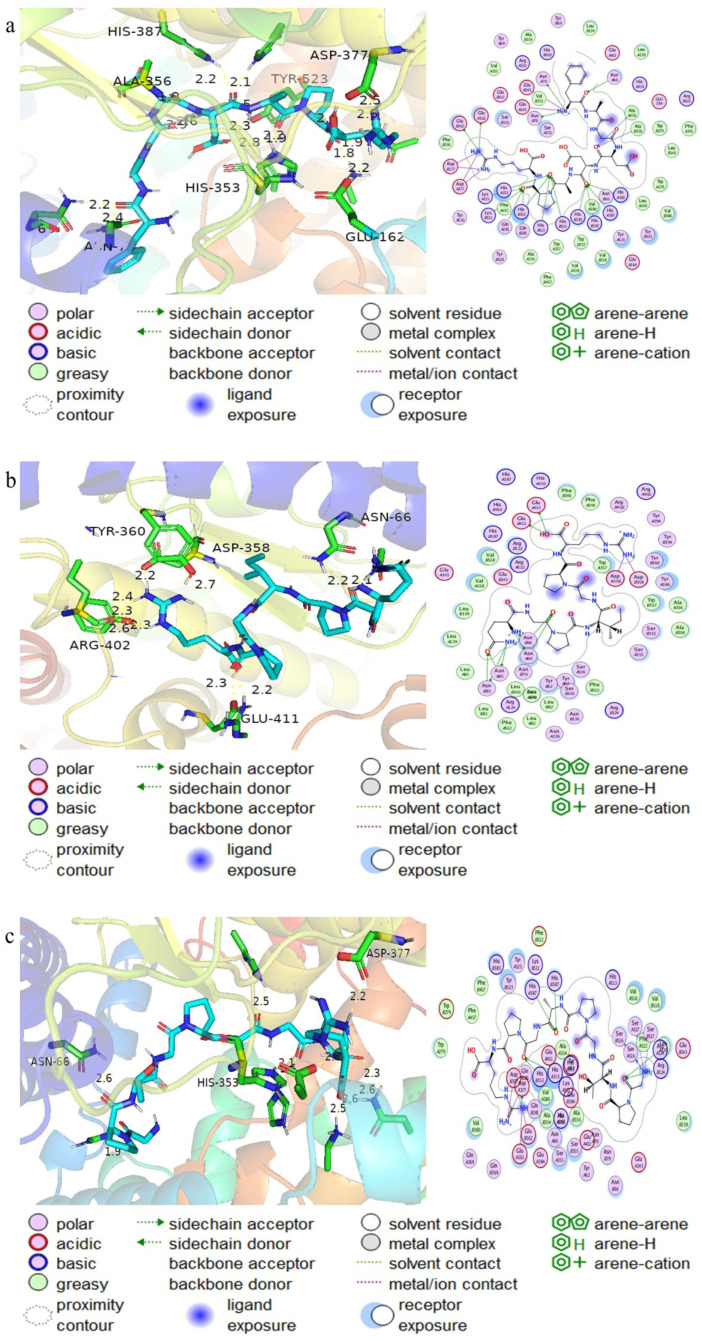
Computational visualization of the optimal docking conformation of the peptide with the ACE active site. On the right, there is a 2D diagram of the interaction between ACE and the substrate, and next to this diagram is a 3D diagram of the hydrophobic surface. (**a**) FAGDDAPR, (**b**) QGPIGPR, and (**c**) GPTGPAGPR.

**Figure 5 biomolecules-14-00581-f005:**
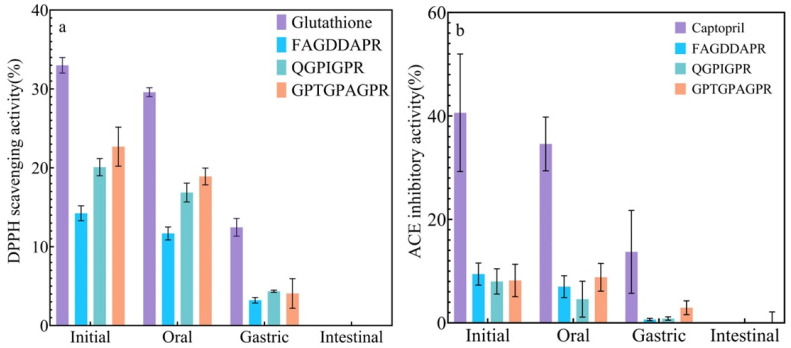
Antioxidant and ACE-inhibitory activities of three peptides were evaluated after simulated digestion. (**a**) DPPH activity of three peptides following oral gastrointestinal simulation; (**b**) ACE-inhibitory activity of six peptides determined after oral gastrointestinal simulation.

**Figure 6 biomolecules-14-00581-f006:**
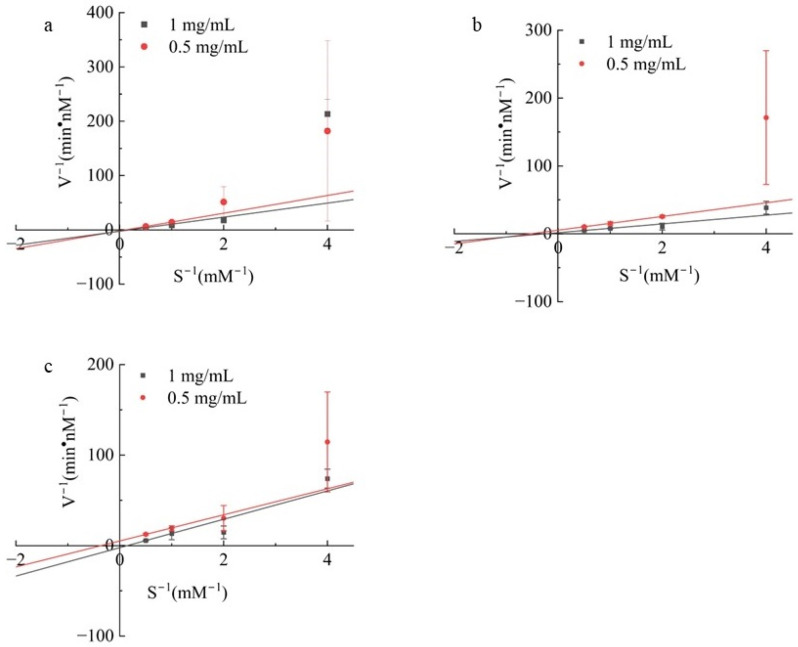
The ACE inhibition patterns of (**a**) FAGDDAPR, (**b**) QGPIGPR, and (**c**) GPTGPAGPR were analyzed using Lineweaver–Burk plots.

**Figure 7 biomolecules-14-00581-f007:**
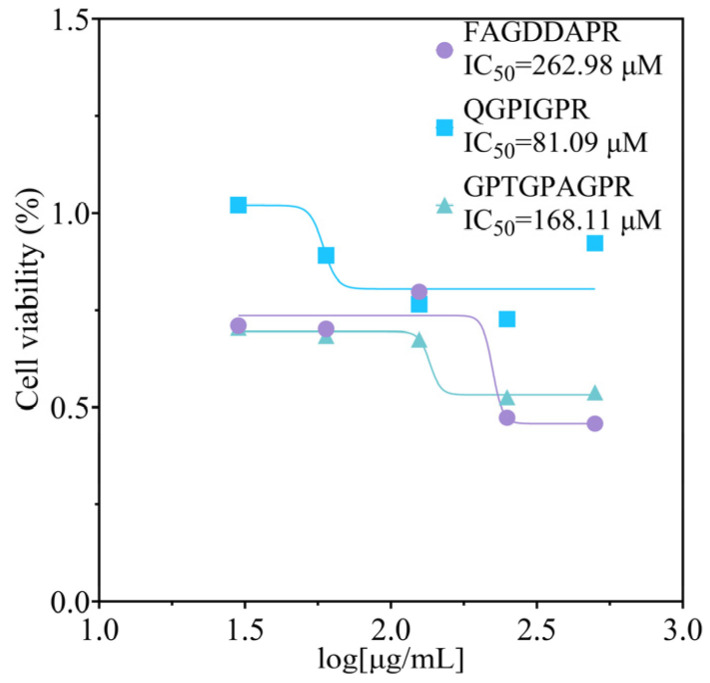
The IC_50_ values of the three peptides.

**Table 1 biomolecules-14-00581-t001:** Results of in silico screening and characterization of different peptides.

Peptide Sequence	FAGDDAPR	QGPIGPR	IFPRNPP	AGFAGDDAPR	GPTGPAGPR
Peptideranker Score	0.641	0.762	0.712	0.562	0.698
AHTpin SVM Score	0.230	0.530	1.450	0.240	0.800
AHTpin prediction	AHT	AHT	AHT	AHT	AHT
Parameter A	0.750	0.857	0.714	0.800	1.111
Parameter B	0.042	0.037	0.036	0.033	0.030
Allergenicity	Non-Allergen	Non-Allergen	Allergen	Allergen	Non-Allergen
Intestinal stability	High	High	High	High	High
Toxicity	Non-Toxic	Non-Toxic	Non-Toxic	Non-Toxic	Non-Toxic
Molecular weight (Da)	847.970	723.930	840.080	976.130	809.010
Half-Life in plasma (s)	845.510	834.410	874.210	878.810	775.810
Hydrophobicity	−0.250	−0.220	−0.180	−0.160	−0.160
Hydropathicity	−0.890	−1.070	−0.790	−0.570	−1.040
Amphiphilicity	0.310	0.530	0.350	0.250	0.270
Isoelectric point	5.756	6.710	6.653	5.802	6.581

**Table 2 biomolecules-14-00581-t002:** Results of molecular docking peptides with ACE.

Name	C Docker Energy	Mode of Action
1o8a_FAGDDAPRR	−10.8	GLU162, GLU376, and SAP377 are involved in facilitating conventional hydrogen-bonding interactions. Additionally, ALA354, GLU162, SAP377, ASN66, ASN70, GLN281, HIS353, ALA356, HIS383, HIS387, LYS511, ARG522, and TYR523 engage in conventional hydrogen-bonding interactions, carbon–hydrogen bonding interactions with HIS387 and HIS513, and Pi-Alkyl interactions with HIS383, PHE457, and TYR523.
1o8a_GPTGPAGP	−10.4	This peptide engages in an attractive charge interaction and forms a salt bridge with GLU162. Additionally, it engages in conventional hydrogen-bonding interactions with ASP377, TYR62, ASN66, ARG124, GLN281, HIS353, HIS383, HIS387, and HIS513. Carbon–hydrogen bonding interactions are established with ALA356, SER516, and SER355, while alkyl interactions occur with LEU139. Finally, pi-alkyl interactions are engaged in with HIS383 and TYR523.
1o8a_QGPIGPR	−9.2	The target protein engages in an attractive charge interaction with ASP358., Furthermore, it engages in conventional hydrogen bonding interactions with TYR62, ASN85, GLU411, TYR394, ARG402, ASP358, ASN66, ASN70, ASN85, and ARG522; alkyl hydrophobic interactions with ALA63 and VAL518; and pi-alkyl interactions with TYR62 and TRP357.

## Data Availability

All data can be made available by the corresponding author on reasonable request.

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
