# Peer review of "Novel Angiotensin-Converting Enzyme-Inhibitory Peptides Obtained from Trichiurus lepturus: Preparation, Identification and Potential Antihypertensive Mechanism"

_biomolecules, 2024, doi:10.3390/biom14050581_

Round 1

Reviewer 1 Report

Comments and Suggestions for Authors

This manuscript describes an interesting set of peptides, derived from ribbonfish, that have ACE inhibitory activity.  Such peptides could have useful properties. 

However, the manuscript is not very clear in a number of areas:

Which proteases were used for hydrolysis and from where were they obtained.  The one which seems to be the best one is alcalase.  From where was it obtained?  Since the peptides are probably derived from hydrolysis in the stomach or gut, why was this one chosen?

Given the number of peptides in the hydrolysis mix, how were the final ones chosen?  Were any of them compared to the known substrates and/or cleavage sites of ACE?  A website was mentioned that was used to predict peptide structure.  What was the point of such prediction?

The peptides were docked into the known structure of ACE.  What was the docking program used?  How were the peptides positioned for docking?  In the figures showing docked peptides, where is the active site?  Where is the metal ion?  It is not clear how the docked peptides relate to each other in the ligand complex structures. This could be clarified with better figures.

How do the docking results compare to the known specificity of ACE?  

What was the purpose of using malvidin and from where was it obtained?  Why was FAPGG used and from where was it obtained?

In general, the manuscript needs careful editing for English usage and grammar.

Comments on the Quality of English Language

The manuscript requires editing for English usage and grammar.

Author Response

Dear Reviewer:

Reviewer 2 Report

Comments and Suggestions for Authors

The authors performed enzymatic hydrolysis of ribbonfish, separated peptides and determined their sequences by LC-MS-MS method. Peptides were identified based on fragmentation spectra. Then, in silico analysis revealed the most interesting compounds, which were subjected to biological assays, including ACE inhibition.

The experimental material is quite interesting and in my opinion suitable for publication. However, the text needs some clarification.

1.            In section 3.3, the authors claim that ". Technical abbreviations are explained when first used". However, the abbreviation FAPGG is not explained.   It may suggest Phe-Ala-Pro-Gly-Gly peptide, whereas it corresponds to N-[3-(2-furyl)acryloyl]-Phe-Gly-Gly ACE substrate.

2. the subtitle of section 3.3 is "LC/MS-MS chromatography and peptide synthesis". Unfortunately, there is no information on peptide synthesis.

3.            When the authors discuss the sequencing of peptides, they provide full information on MS/MS data acquisition, but I did not find how the sequence is actually determined (Sequencing de novo? Sequencing from genomic data? How was the sequence verified?)

4.            It is not clear to me whether peptides were synthesized. If so, what was the purpose of the synthesis?

The synthetic peptides could be used to confirm the data obtained by sequencing.  Also, a comparison of the biological activity of natural samples isolated from digest and synthetic peptides would be advantageous.

5.            Currently there are several examples of peptides with ACE inhibitory properties. Please compare the activity and other properties of the identified peptides with previously reported compounds.

Author Response

Dear Reviewer:

Reviewer 3 Report

Comments and Suggestions for Authors

The presented manuscript is devoted to the characterization of new peptides with inhibitory activity against ACE isolated from an unconventional source. It should be said that the idea sounds quite reasonable and seems interesting. The structure of the experiments, or at least the presentation of the materials and methods, also seems quite reasonable. However, familiarization with the Results and Discussion sections causes some disappointment.

First of all, I want to note the extremely careless design of the manuscript. There are a lot of typos in it, "someone else's" text that does not fit into the logic of the narrative (see comments below), a lot of either unnecessary punctuation marks or omitted ones. The logic of the presentation of the material is violated.

My specific comments are as follows:

Lines 35-39: I suggest that the authors re-read what is written here and correct the obvious typos.

Line 225: Why were these three peptides carefully selected? The authors should explain their choice and selection criteria. In my opinion, the logic of the text is broken here. First, you need to explain the choice by presenting the actual results (in a table or in figures), and only then inform that the authors will continue to work with the specific peptides they have chosen.

Lines 226-228: What is it? Why did these 2 phrases appear here?

Lines 231-236: What kind of research are we talking about? About the results obtained in the course of this work? It would be more correct to provide some data, rather than a descriptive part.

Line 243: Table 1 shows the corresponding results in silicon. In silicon????

Figure 5 caption: Have three or six peptides been analyzed?

Section 3.7: In my opinion, the authors should carefully analyze the nature of the graphs and, according to textbooks on enzymatic kinetics and inhibitory analysis, discuss by what mechanism each of the peptides inhibits ACE. It's not enough to just give links to other people's work. A full-fledged analysis is needed.

More extensive Discussion is needed.

Comments on the Quality of English Language

English should be carefully corrected by a native speaker.

Author Response

Dear Reviewer:

Round 2

Reviewer 1 Report

Comments and Suggestions for Authors

The authors have addressed all of the concerns.

Author Response

The authors thank the reviewer's comments.

Reviewer 3 Report

Comments and Suggestions for Authors

Dear Authors,

The current version of the manuscript looks more appropriate. However, there are still a lot of typos all over the text. Please check the text carefully. 

Comments on the Quality of English Language

English must be checked by native speaker.

Author Response

Please find attached revised manuscript
